# Creating Custom Neural Circuits on Multiple Electrode Arrays Utilizing Optical Tweezers for Precise Nerve Cell Placement

**DOI:** 10.3390/mps3020044

**Published:** 2020-06-20

**Authors:** Frank H. Kung, Ellen Townes-Anderson

**Affiliations:** 1Department of Cell Biology and Neuroscience, Rutgers University, Piscataway, NJ 08854, USA; 2Department of Pharmacology, Physiology, and Neuroscience, Rutgers University-New Jersey Medical School, Newark, NJ 07103, USA; andersel@njms.rutgers.edu

**Keywords:** optical tweezers, multiple electrode array, electrophysiology, cell patterning, synaptic plasticity

## Abstract

Precise creation, maintenance, and monitoring of neuronal circuits would facilitate the investigation of subjects such as neuronal development or synaptic plasticity, or assist in the development of neuronal prosthetics. Here we present a method to precisely control the placement of multiple types of neuronal retinal cells onto a commercially available multiple electrode array (MEA), using custom-built optical tweezers. We prepared the MEAs by coating a portion of the MEA with a non-adhesive substrate (Poly (2-hydroxyethyl methacrylate)), and the electrodes with an adhesive cell growth substrate. We then dissociated the retina of adult tiger salamanders, plated them onto prepared MEAs, and utilized the optical tweezers to create retinal circuitry mimicking in vivo connections. In our hands, the optical tweezers moved ~75% of photoreceptors, bipolar cells, and multipolar cells, an average of ~2000 micrometers, at a speed of ~16 micrometers/second. These retinal circuits were maintained in vitro for seven days. We confirmed electrophysiological activity by stimulating the photoreceptors with the MEA and measuring their response with calcium imaging. In conclusion, we have developed a method of utilizing optical tweezers in conjunction with MEAs that allows for the design and maintenance of custom neural circuits for functional analysis.

## 1. Introduction

Understanding the function of individual neurons has increased our understanding of the mechanisms behind synaptic plasticity [1], which is the basis for learning and memory. In addition, this understanding can enhance our potential for developing effective neuroprosthetics [2]. To examine individual neurons, electrophysiological methods such as patch clamping allow researchers to stimulate and record from individual neurons [3]. However, patch clamp techniques are limited in their ability to measure signals from or stimulate cells over long periods of time and are limited in the number of cells that can be simultaneously stimulated and analyzed [3].

One technology designed to overcome these limitations is the multiple electrode array (MEA). An MEA is generally composed of a number of small (10–30 µm) flat electrode heads embedded in a substrate such as glass. These can be encased in a plastic or polymer dish, allowing researchers to maintain sterility of a cell culture over long periods of time. Without specific techniques, cells cultured on MEAs create random patterns of cells on the electrodes, which, while useful in many applications, does not allow for the measurement of individual neurons. Moreover, multiple cells can cover a single electrode. Therefore, stimulation of a culture via MEAs will often activate several different cellular networks, even when using a cell-selective means of activation [4].

Some methods exist that allow researchers to place single neurons on electrodes, but they are often cumbersome and technically difficult. For example, one can pattern a substrate that the cells require for attachment onto the electrodes, thus limiting the potential area of attachment for each neuron to a specific electrode [5]. However, substrate patterning requires precise alignment with the small electrodes [5]. In addition, in order to place multiple types of neurons onto specific electrodes, one must design custom coatings for each electrode [6].

Optical tweezing is a technique that utilizes a low-powered laser and was originally used to manipulate subcellular components [7]. We have successfully used the tweezers to physically manipulate individual cells [8,9,10]. Our technique is compatible with many different substrates and allows the user to place specific neurons in specific locations, without damage to the neuron or the substrate. Using optical tweezers protocols developed by us, one group was able to place individual embryonic hippocampal granule cells onto a custom MEA with a “neurocage” design [11]. Furthermore, they confirmed our original findings that the 980 nm laser can be safely used to optically trap neurons.

The present study further expands the optical tweezers technique, to test electrophysiological approaches to the study of neural networks. We utilized a commercially available MEA, along with an optical tweezing system, to arrange all types of adult retinal neurons into potential neural circuits. The tweezed cells grew successfully in culture and responded to electrical stimulation.

## 2. Materials and Methods

### 2.1. Animals

Retinal tissue and cells were obtained from adult aquatic-phase tiger salamanders (*Ambystoma tigrinum*, 18–23 cm in length), maintained at 5 °C, on a 12-h light/12-h dark cycle. Animals were adapted to the light cycle for at least 1 week before use. All protocols were approved by the Institutional Animal Care and Use Committee at Rutgers New Jersey Medical School.

### 2.2. Retinal Dissociation and Cell Identification

Salamander Ringer (85 mM NaCl, 1.5 mM KCl, 25 mM NaHCO_3_, 0.5 mM CaCl_2_, 0.5 mM NaH_2_PO_4_, 24 mM glucose, 0.03 mM phenol red, 1.0 mM sodium pyruvate) and salamander media (108 mM NaCl, 2.5 mM KCl, 2 mM HEPES, 1 mM NaHCO_3_, 1.8 mM CaCl_2_, 0.5 mM NaH_2_PO_4_, 1 mM NaHCO_3_, 24 mM glucose, 0.5 mM MgCl_2_, 1 mM Na pyruvate, 7% medium 199, 1× minimum essential (MEM) vitamin mix, 0.1× MEM essential amino acids, 0.1× MEM nonessential amino acids, 2 mM glutamine, 2 μg/mL bovine insulin, 1 μg/mL transferrin, 5 mM taurine, 0.8 μg/mL thyroxin, 10 μg/mL gentamicin, and 1 mg/mL bovine serum albumin (pH 7.7)) were prepared as previously described [12]. Salamander retinas were isolated as previously described, with some modification [12]. Briefly, salamanders were enucleated, and the anterior portion of the eyes was removed. Each eyecup was then bisected, using a razor blade. Each half retina was then removed from each half eyecup, using curved forceps, washed 2 times with salamander Ringer, and kept in salamander Ringer at 10 °C, until dissociation. Dissociation was performed immediately prior to optical tweezing, by placing each retinal piece into a 1.5 mL papain solution consisting of 14 U/mL papain (Worthington, Freehold, NJ, USA) in salamander Ringer’s solution, for 40 min, at room temperature. Each retinal piece was then washed twice with salamander Ringer and finally gently triturated in 1 mL of ice-cold salamander Ringer. Then, 50 µL of cell suspension was immediately used for optical tweezing.

Cells were identified by using their morphology, as previously described [13,14] (see Figure 4). For retinal neurons, many of the five cell types can be reliably identified by their distinguishing axonal and dendritic extensions. All salamander cells are large. Photoreceptor cell somas measure about 10–15 µm in diameter. In addition, photoreceptors have an extension of the soma called the inner segment, which contains a collection of mitochondria called an ellipsoid. Rod cells have a triangular ellipsoid, whereas cone photoreceptors contain a spherical ellipsoid. Rod cells may also have a long rod-like outer segment, the photosensitive part of the sensory cell, and an axon terminal extending from a short axon-like process; however, during dissociation, these structures can become separated from the cell body of the rod photoreceptor. In contrast, cone photoreceptors have an axon terminal directly attached to the cell body and a short cone outer segment, which is usually not retained during dissociation. Bipolar cells can be identified based on the presence of a Landolt club, a thick primary dendrite; moreover, their cell somas are slightly smaller, about 8–10 µm. Horizontal, amacrine, and ganglion cells have distinctive dendritic trees with multiple branches emerging from the cell soma. However, it was uncommon for the branching neuritic processes to be adequately retained after retinal dissociation, and therefore these cells were grouped into a “multipolar” category. On occasion, however, ganglion cells were recognizable because of their large cell bodies, ~20 µm, prominent primary dendrite, and thin single axon.

### 2.3. MEA Coating and Preparation

Multiple Electrode Arrays (MEAs) were purchased from Multi Channel Systems MCS GmbH (Reutlingen, Germany). Arrays chosen for these experiments had an electrode diameter of 10 µm and an interelectrode distance of 30 µm (Multi Channel Systems MCS GmbH, Cat #60ThinMEA30/10iR-ITO-(w/o)). The titanium nitride (TiN) electrode heads with conductive strips of transparent indium tinoxide (ITO), along with silicon nitride (SiN) insulation isolating each electrode, were fabricated on top of #1 coverslip glass (0.13–0.17 mm thick). The total thickness of the MEA arrays is 180 µm. It was necessary to use thin MEAs due to the optics of the objective used for optical tweezing (a 40x oil immersion plan neofluor objective with a numerical aperture of 1.3 and a working distance of 210 µm), as well as to focus the laser on a single cell. No physical alteration of these MEAs was required. These arrays were cleaned overnight in a 1% *w/v* Tergazyme (Sigma Aldrich, Cat# Z273287, St Louis, MO) solution, then washed three times with DI H2O, and finally sterilized under UV, in a biosafety hood, for 1 h.

Poly (2-hydroxyethyl methacrylate) (p-HEMA) (Sigma Cat#3932) solution was prepared as previously described, by dissolving 20 mg/mL of p-HEMA in 95% ethanol (EtOH), overnight, rocking at RT [9,10]. A p-HEMA coating was necessary to prevent adhesion of retinal cells to some areas of the MEA so that the optical tweezers could pick up and move the retinal cells. This solution was applied to specific areas of the surface of the MEA by placing the MEA in a 35 mm dish, such that it was inclined at a 60° angle. Then, 100 µL of p-HEMA solution was carefully dripped onto the surface of the MEA, being careful to not allow the p-HEMA solution to cover the central electrode region (Figure 1). The MEAs were then laid flat into 94 mm dishes, covered, and allowed to dry for 1 h in a biosafety hood. The MEA was then rotated 90°, and the p-HEMA coating was repeated so that ¾ of the total surface area of the MEA was coated, but not the electrodes in the center (Figure 1). If p-HEMA did accidentally drip onto the electrode area, the MEA was quickly sprayed with 70% EtOH, washed 3 times with sterile DI H2O, and allowed to dry. The MEA was then recoated with p-HEMA, using the steps previously described.

Polydimethyl siloxane (PDMS) rings were made to hold liquid on the MEA. PDMS (Dow Corning Corporation Cat#3097358-1004) was made as previously described [15]. Briefly, elastomer base was vigorously mixed with the curing agent in a 10:1 ratio by weight. The solution was then placed in a desiccator, under vacuum, for 30 min, to remove air bubbles. The PDMS polymer was poured into a 94 mm culture dish, placed under vacuum for another 30 min, and finally cured in a 70 °C oven for at least 2 h. A ring with a 1” outer and ¾” inner diameter was punched from the PDMS slab, cleaned using Scotch Tape, and sterilized by submerging in 70% EtOH. Subsequently, the PDMS ring was washed with sterile DI H_2_O twice and allowed to dry under UV light overnight. The PDMS ring was then placed on the MEA around the central electrodes. Vaseline was applied around the outside of the PDMS ring, in order to ensure there would be no leakage of media during culture.

The ¼ of the MEA not covered with p-HEMA was coated with a substrate that provides adhesion for salamander neurons, a monoclonal antibody called Sal-1 [16]. First, 75 µL of goat-anti-mouse IgG (0.1 mg/mL in salamander Ringer) was placed inside the ring for 3 h; the goat-anti-mouse IgG solution was then aspirated off of the MEA, washed twice with 75 µL of salamander Ringer, and finally coated with 75 µL Sal-1, overnight, at 10 °C. Care was taken during the coating and washing steps to ensure no liquid spilled onto the p-HEMA surface. Immediately prior to optical tweezing, the Sal-1 was aspirated and the MEA washed twice with salamander Ringer before filling the culture area with 1 mL of ice-cold salamander media. The MEA was placed into a sterile 100 mM Petri dish.

To reuse MEAs, PDMS rings were removed and thrown away. Vaseline was removed, using cotton swabs soaked in isopropyl alcohol. The MEA was then briefly dipped in 70% EtOH, washed 3 times in DI H_2_O, and finally soaked in 1% Tergazyme solution, overnight, before being extensively washed with DI H_2_O and stored in DI H_2_O.

### 2.4. Optical Tweezing

Optical tweezing was performed in a similar manner to that previously described [8,9,10]. Briefly, an 800 mw 980 nm continuous wave diode laser (Cell Robotics Inc., Albuquerque, NM, USA) was mounted onto an Axiovert 100 inverted light microscope (Carl Zeiss Inc., Oberkochen, Germany) with a television port. Laser light was directed through a custom-drilled hole in the bottom of the microscope, allowing for an optical path through the objective to the cells. The microscope was equipped with a motorized stage (Cell Robotics Inc.), and both the laser and stage were controlled by a custom software package based on Labview (National Instruments, Austin, TX, USA) from Bloomy Controls (Windsor, CT, USA). Laser intensity was controlled through an NI DAQ UMI7764 controller (National Instruments), and stage control and motion control feedback were connected to the NI DAQ through a motor control unit MCDC 3006S (Faulhaber, Schönaich, Germany). For optical tweezing, a 40× oil immersion plan neofluor objective with a numerical aperture of 1.3 was used with brightfield optics.

A prepared MEA was placed upon the stage, and the central electrode array area was set as the home reference point. Then, 50 µL of cell suspension was pipetted onto the area covered by p-HEMA, using a 1000 µL tip. The MEA was covered with a sterile 35 mM Petri dish cover, to maintain sterility. The p-HEMA area was then searched for retinal cells. Cells chosen for tweezing were those that appeared “unattached” to the p-HEMA surface, determined by gently tapping the microscope stage and observing the reaction of the cells. Identifiable cells were imaged, and their initial location was recorded. A cell was then picked up by the laser beam, by focusing slightly above the cell, setting the optical tweezers to 40–100% power (depending upon the difficulty in picking up the cell), and gradually focusing upward from the surface of the MEA. The power was then lowered to 25%, and the cell, thus “optically trapped”, was moved toward the central electrodes by moving the stage. At the electrodes, the power was lowered again to 10%, and the cell slowly maneuvered over an electrode. After maneuvering, laser power was lowered to 6%, and the focus was brought closer to the MEA electrodes (see Appendix A). Finally, the laser was turned off, and the cell was allowed to gently settle onto the electrode. As shown in Appendix A, care was taken to minimize movement over the ITO tracks and to prevent focusing the laser directly upon the MEA electrodes, as these actions could damage the electrodes and insulation. When directly focused on the electrodes or tracks, the laser beam, at high power, was absorbed and caused significant damage to both. The final location of each cell was logged and compared to the starting location. Success of movement of cells was defined as being able to pick up a cell, move it to an electrode, and place it onto an electrode. One-third or more of the electrode must also have been covered by the cell. After cells were plated and moved, they were maintained at 10 °C. Cells were allowed to grow for 1 week and imaged daily, using brightfield optics on an Axiovert 135 (Carl Zeiss Inc., Oberkochen, Germany).

### 2.5. Ca^2+^ Imaging and Stimulation

Calcium levels in optically tweezed retinal cells were measured, using Fura-2 AM (Life Technologies, Cat# F1221, Carlsbad, CA, USA). Then, 5 µL of 1 µg/µL Fura-2 in DMSO was added to 1 µL of 20% pluronic acid (Molecular Probes, P-6867, Eugene, OR, USA) and briefly vortexed. A total of 1 mL of salamander Ringer was then added and vortexed for 5 min. After rinsing the MEA with cells in salamander Ringer for 5 min, 1 mL of Fura-2 containing solution was added. Cultures were incubated at RT for 40 min, while gently rocking. Fura-2 solution was then carefully aspirated off, and the MEA was washed with 1 mL of salamander Ringer, for 30 min, at RT, while rocking. Fura-2 was excited at 340 and 380 nm, using a Polychrome V (Till Photonics, Bavaria, Germany) monochromator coupled to an inverted Leica DM IRB microscope. Image acquisition was accomplished with an Imago CCD camera (Till Photonics) controlled by an Image Control Unit (Till Photonics). Cells were exposed for 50 ms to each wavelength and imaged every second. Signals were averaged over the entire cell not covered by the electrode.

Stimulation was accomplished with an MEA1060 preamplifier (Multi Channel Systems MCS GmbH) with an attached STG4002 stimulator (Multi Channel Systems MCS GmbH). Parameters for stimulation were set by using the MC Stimulus II software provided by Multi Channel Systems MCS GmbH. We used two different types of stimulation: rectangular and sinusoidal. Rectangular stimulations consisted of 100 µs negative stimulation, followed by 60 µs of off, followed by 100 µs of positive stimulation. The sinusoidal stimulation frequency was set to 5 Hz. For each type of stimulation, we used three amplitudes, −100, −200, and −400 mV. In addition, we used a −1 µA current stimulation for sinusoidal stimulation. All electrodes not covered by cells were connected to ground. Prior to stimulation, cells were imaged for 50 s, to establish a baseline calcium reading. After stimulation, cells were allowed to recover, to return to baseline, before testing other stimulation parameters. Cells which did not display any increase over their baseline response at −400 mV rectangular or −1 µA sinusoidal stimulation were not analyzed. The number of unstimulated rod photoreceptors was not quantified, as it was difficult to determine if the lack of stimulation was due to an MEA defect, slight movement of the cells off of the electrode, incomplete penetration of Fura-2 AM into the cell, or lack of response by the cell to extracellular stimulation. Significance was achieved at *p* < 0.05, as determined by Student’s *t*-test.

Stimulation was accomplished with an MEA1060 preamplifier (Multi Channel Systems MCS GmbH) with an attached STG4002 stimulator (Multi Channel Systems MCS GmbH). Parameters for stimulation were set by using the MC Stimulus II software provided by Multi Channel Systems MCS GmbH. We used two different types of stimulation: rectangular and sinusoidal. Rectangular stimulations consisted of 100 µs negative stimulation, followed by 60 µs of off, followed by 100 µs of positive stimulation. The sinusoidal stimulation frequency was set to 5 Hz. For each type of stimulation, we used three amplitudes, −100, −200, and −400 mV. In addition, we used a −1 µA current stimulation for sinusoidal stimulation. All electrodes not covered by cells were connected to ground. Prior to stimulation, cells were imaged for 50 s, to establish a baseline calcium reading. After stimulation, cells were allowed to recover, to return to baseline, before testing other stimulation parameters. Cells which did not display any increase over their baseline response at −400 mV rectangular or −1 µA sinusoidal stimulation were not analyzed. The number of unstimulated rod photoreceptors was not quantified, as it was difficult to determine if the lack of stimulation was due to an MEA defect, slight movement of the cells off of the electrode, incomplete penetration of Fura-2 AM into the cell, or lack of response by the cell to extracellular stimulation. Significance was achieved at *p* < 0.05, as determined by Student’s *t*-test.

## 3. Results

### 3.1. Coating and Movement of Cells

Dissociation of salamander retina yields cells from all the layers of the retina. This includes rod and cone photoreceptors, bipolar cells, horizontal, amacrine and ganglion cells, grouped here as multipolar cells, and Mueller glia. A description of these cells and their morphologies after isolation is described in more detail elsewhere [13,14] and in the Methods section. All types of retinal cells could be moved onto electrodes via optical tweezing. This included rod photoreceptors with or without their outer segments or axon terminals. For multipolar cells, we used only those with at least one neurite still attached. Individual ganglion cells were occasionally identified and also successfully moved.

Success of movement of cells was defined as being able to pick up a cell, move it to an electrode, and place it onto an electrode, covering a third or more of the surface of the electrode. Cells were moved an average distance of ~2 mm, with a maximum distance of ~4 mm (Figure 2). Of cells selected for movement, 75% were successfully placed on electrodes. No statistical differences between the different cell types were observed for either success rate or total amount of movement (Figure 3). Cells which were not successfully placed on electrodes were typically “lost” in the cell medium due to optical tweezing. It was unclear what happened to these cells and whether they remained viable or not. Floating cells and cell parts were removed when media was changed, but some cell debris did fall and attach to the Sal-1 substrate. In our hands, optical tweezing moved cells at a speed of ~16–17 microns/second (data not shown). The lifting up of cells prior to movement and placing of cells onto the electrode after optical tweezing was a relatively quick process and took only ~30 s.

### 3.2. Culture and Growth of Cells

All retinal cell types survive, sprout processes, and grow neurites in vitro after being placed onto MEA electrodes, in a manner similar to cells not grown on electrodes [13,14] or subject to optical tweezing [8,9,10]. Due to the presence of the electrodes, which partially block the cell morphology, the growth of processes on the MEA was not quantified. Adult retinal cells did not migrate off the electrodes (Figure 4), unlike embryonic granule cells [11], but their position did change slightly as they flattened and grew processes.

During their growth, retinal cells form presynaptic varicosities, which are large swellings on neurites containing synaptic vesicles and other synaptic proteins. These presynaptic specializations were occasionally seen to contact other cells, indicating that synaptic connections were being developed with other cells (Figure 5A, arrows indicate the base of a varicosity as it extends to a cone cell). Previous research from our lab has shown that the frequency of formation of these varicosities is dependent upon the identity of the neighboring cells [9]. However, neuritic contacts were observed between photoreceptors and all other cell types, including both types of photoreceptors, bipolar, and multipolar cells. Figure 5A demonstrates the formation of a simple network after seven days in vitro (DIV) amongst three photoreceptors, two rod cells and one cone cell. These networks were also created by using a wider variety of cells by patterning photoreceptors, bipolars, and multipolars onto specific areas (Figure 5B).

### 3.3. Activity and Calcium Response of Cells

Rod and cone photoreceptors and bipolar cells do not have action potentials and instead rely on graded membrane potentials for their function. Of the multipolar retinal neurons (horizontal cells, amacrine cells, and ganglion cells), only ganglion cells have been described to have spontaneous action potentials [14]. Ganglion cells make up only a small portion of the retinal culture, and, thus, electrical activity was not recorded from the cultures.

In order to assess activity of individual retinal cells placed on specific electrodes, we used calcium imaging. Using a calcium-sensitive dye, Fura-2, we could image and analyze changes in levels of calcium due to the opening of voltage-sensitive calcium channels. We focused on the Ca2+ response of rod photoreceptors, using several types of stimulation. The first was rectangular stimulation, as described in the Methods section. This stimulation hypothetically results in a general depolarization of the cell membrane, causing a voltage-gated calcium response [17,18]. We also used a low-frequency sinusoidal stimulation. This stimulation hypothetically results in specifically activating L-type calcium channels, which are expressed on photoreceptor axon terminals [19]. By examining these two types of stimulations, we could examine physiologically relevant stimulations which are currently being researched as potential extracellular stimulation parameters for retinal prostheses [18].

The response to stimulation did not depend upon whether the ellipsoid or the nuclear end of the cell covered the stimulating electrode. Moreover, rod photoreceptors with and without axons or outer segments demonstrated similar Ca2+ responses to extracellular electrode stimulation. One day after tweezing, rod photoreceptors demonstrated responses to both rectangular and sinusoidal stimulations. The lowest threshold for rectangular stimulation was −200 and −400 mV for sinusoidal stimulations. The greatest calcium response was to sinusoidal stimulations of -1 µA (Figure 6).

After seven DIV, the same rod photoreceptors were again stimulated, using the same parameters. Cells had remained on the electrodes during this time, and no significant cell death was seen. Rod-cell responses to rectangular stimulation were similar to those at day one; however, their responses to −400 mV and −1 µA sinusoidal stimulation had significantly decreased (Figure 6).

## 4. Discussion

The protocol that we have described, utilizing optical tweezers in conjunction with an MEA, allows a user to specifically maneuver individual neurons onto selected electrodes. The substrate used here to make the MEA non-adhesive (p-HEMA) so that cells can be picked up by the laser should have general applicability. However, the adhesive substrate used for cell growth is specific for adult salamander neurons and, thus, should be selected depending on the cell system used. Other cell-placement techniques, such as cell printing [20,21], have been suggested. Nakamura et al., in 2005, demonstrated that they could “print” dots of cells with an accuracy of 8.8 to 17.6 µm. However, each dot varied in the number of cells it contained and was 80 to 260 µm in diameter, making it unsuitable for the purpose of printing onto 10 µm electrodes [21]. Thus, because of the micromanipulation, our method is highly accurate for single cell placement and a significant improvement over previous techniques.

Our method is compatible with a commercially available MEA system. Optical tweezers are also commercially available from several companies or can be built using widely available instructions [22,23]. The optical tweezers are robust enough to work on cells without a spherical shape, as demonstrated by both the pioneering work of Arthur Ashkin [24] and our use of amphibian rod and cone photoreceptors, and bipolar and multipolar cells, some of which have a total length exceeding 50 µm. Previously, our group and others have also been able to move mammalian cells (data not shown, [11]). Measuring the calcium response allowed us to validate the effectiveness of the extracellular electrodes in stimulating micromanipulated rod photoreceptors in vitro. Specifically, rectangular stimulation was able to cause a significant calcium response; current gated low-frequency sinusoidal stimulations were greater in comparison to either rectangular or voltage gated sinusoidal responses. After seven DIV, the threshold of amplitude required to get a Ca2+ response increased. It is possible that this was due to the increase in membrane from growth in vitro so that the amount of charge required to activate the channels was larger compared to rod photoreceptors at 1 DIV. We suggest that activity could also potentially be recorded with voltage-sensitive dyes or through the electrode, depending on the magnitude of the activity.

In the future, researchers may be able to take advantage of the specific nerve cell placement with optical tweezers to examine a variety of issues regarding synaptogenesis and/or regenerative growth. For instance, placing neurons onto specific electrodes within a single culture dish and supplying a variety of target nerve cells could test synaptic selectivity of the presynaptic neurons. Placement of adjacent glial cells could allow examination of the role of glial in synaptic formation or regenerative growth. In addition, stimulatory protocols could be tested on individual neurons to learn about the effect of activity on synaptogenesis and growth. Micromanipulation of neurons also allows networks to be formed with precision, and the MEA allows functional analysis of these networks over time. Finally, the testing of various types of stimulation may help develop effective neural protheses for specific nerve-cell types or circuits.

Setups that allow for multiple optical tweezer traps have been utilized to manipulate other cell types and may further increase the speed of the placement of cells onto the electrodes, whereas it may be possible to integrate a nanophotonic optical tweezer array with an MEA and eliminate the need for bulky optical tweezers assemblies [25,26]. However, it may be difficult to accurately identify, trap, and position multiple different types of cells based on morphology with these approaches.

The limitations of our technique include the concern that some neurons may be sensitive to the 980 nm laser light used for tweezing. In our hands, only retinal pigmented epithelial cells with melanin granules absorbed damaging amounts of laser power [27].

## Figures and Tables

**Figure 1 mps-03-00044-f001:**
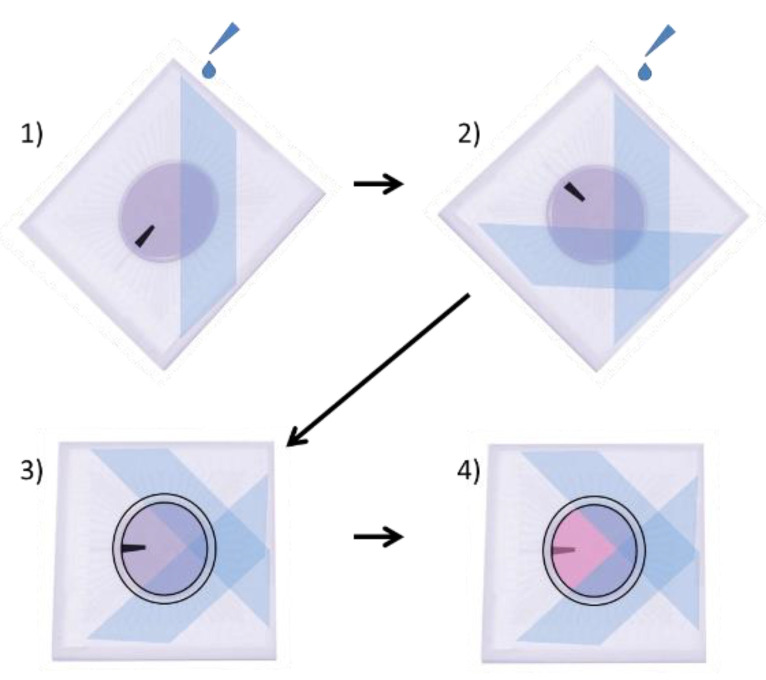
Procedure for coating MEAs with p-HEMA. (**1**) MEA is balanced at a 60° angle in a 35 mm dish, and 100 µL of p-HEMA is dripped onto the MEA, taking care to not allow the electrodes in the center of the MEA to be covered with p-HEMA. The MEA is then covered in a 90 mm dish and allowed to dry for 1 h. (**2**) The MEA is turned 90°, and p-HEMA is again placed at a 60° angle in a 35 mm dish, and 100 µL of p-HEMA is dripped onto the MEA, again taking care not to allow p-HEMA to drip onto the center electrodes. The MEA is once again covered in a 90 mm dish and allowed to dry for 1 h. (**3**) A PDMS ring is applied to the MEA, and Vaseline is applied around the ring, to prevent leakage of media. (**4**) The MEA is coated with 75 µL of Sal-1. (The black bar is provided to help visualize changes in orientation.).

**Figure 2 mps-03-00044-f002:**
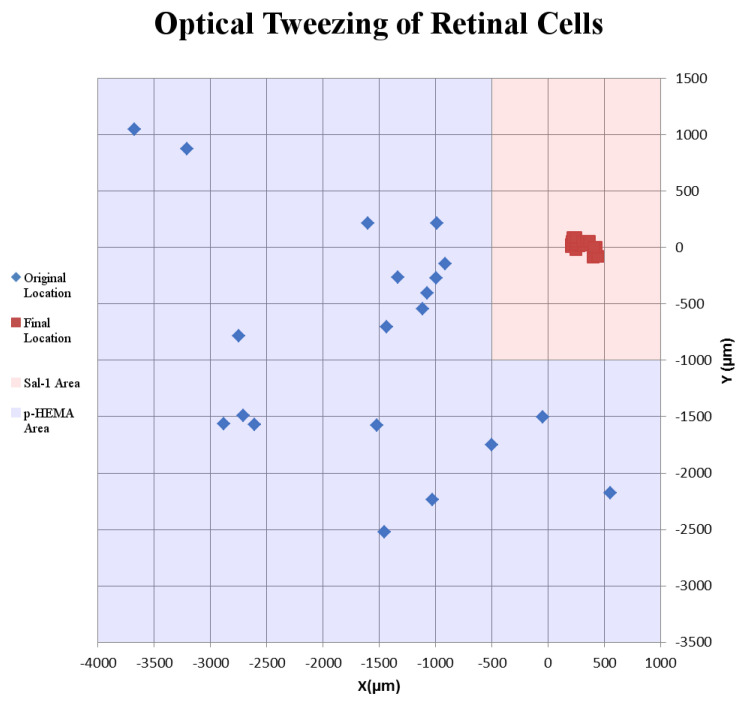
Example of the movement of retinal cells on a microelectrode array. Blue diamonds indicate original location where retinal cells were found in the p-HEMA-coated area. Red squares indicate final location of retinal cells tweezed onto MEA electrodes. Blue shaded squares indicate p-HEMA coating. Red shaded regions indicate Sal-1-coated areas.

**Figure 3 mps-03-00044-f003:**
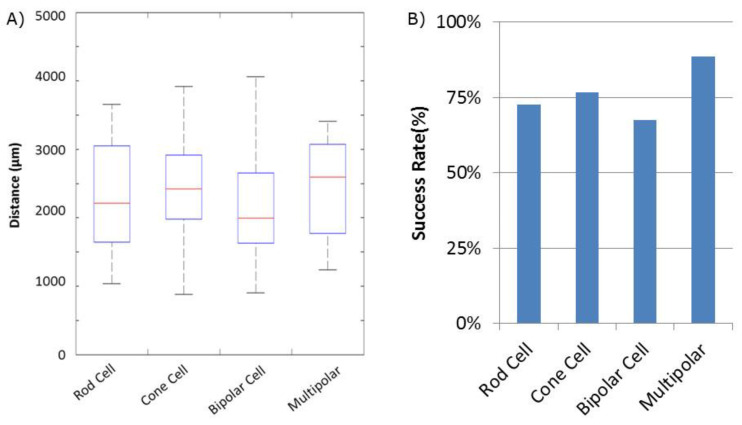
Quantification of retinal cell movement during experiments. All cell types, except for Mueller cells, were moved. (**A**) Distance each cell type was tweezed. Average distance indicated by the Red Line. Blue box indicates 25% quartile. Dashed lines with end caps indicate shortest and longest distance each cell type was moved. (**B**) Success rate of movement for each cell type. Success was defined as identifying a moveable cell and being able to place it such that some part of the cell was on an MEA electrode.

**Figure 4 mps-03-00044-f004:**
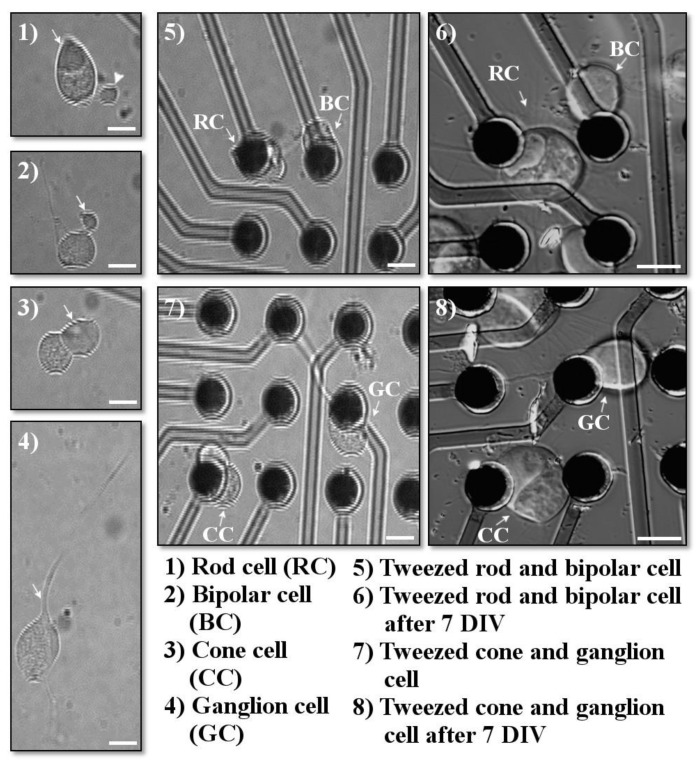
Examples of cells chosen for optical tweezing. Cells were imaged immediately prior to tweezing, immediately after tweezing, and after several days in vitro. (**1**) Arrow points to the triangular ellipsoid typical for rod cells; arrowhead points to the rod axon terminal. (**2**) Arrow indicates the bipolar Landolt club. (**3**) Arrow points to the spherical cone ellipsoid. (**4**) Arrow indicates the ganglionic primary dendrite. (**5**–**8**) Retinal cells after optical tweezing (**5**,**7**) immeditely after tweezing; (**6**,**8**) 7 days in vitro (DIV) after tweezing. Scale bars = 10 µm.

**Figure 5 mps-03-00044-f005:**
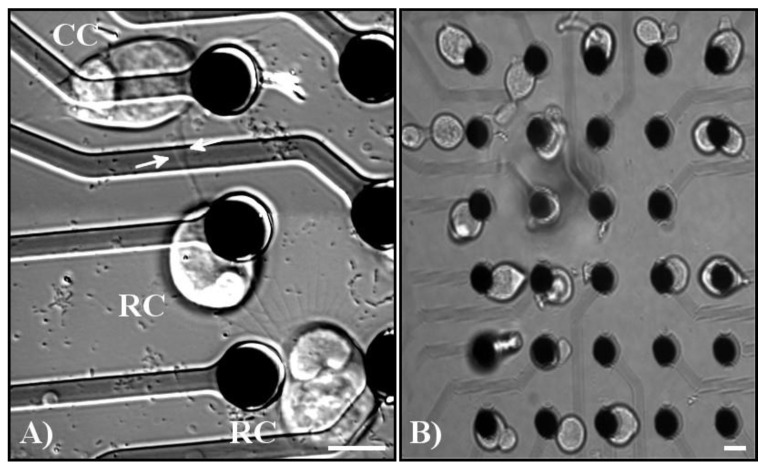
(**A**) Example of varicosity formation and network formation amongst tweezed cells. Pictured from top to bottom: cone photoreceptor (CC), rod photoreceptor (RC), and a rod photoreceptor. White arrows indicate the base of a varicosity extending from the middle rod photoreceptor to the neighboring cone photoreceptor. All three cells contacted each other through processes formed by the cells. (**B**) Example of groups of cells tweezed onto MEA electrodes demonstrating the precision that can be accomplished by using the MEA. From left to right, the first column of electrodes contains rod and cone photoreceptors, the second column of electrodes contains bipolar cells, the third column of electrodes contains multipolar cells, the fourth column contains bipolar cells, and the fifth column contains photoreceptors. Scale bars = 10 µm.

**Figure 6 mps-03-00044-f006:**
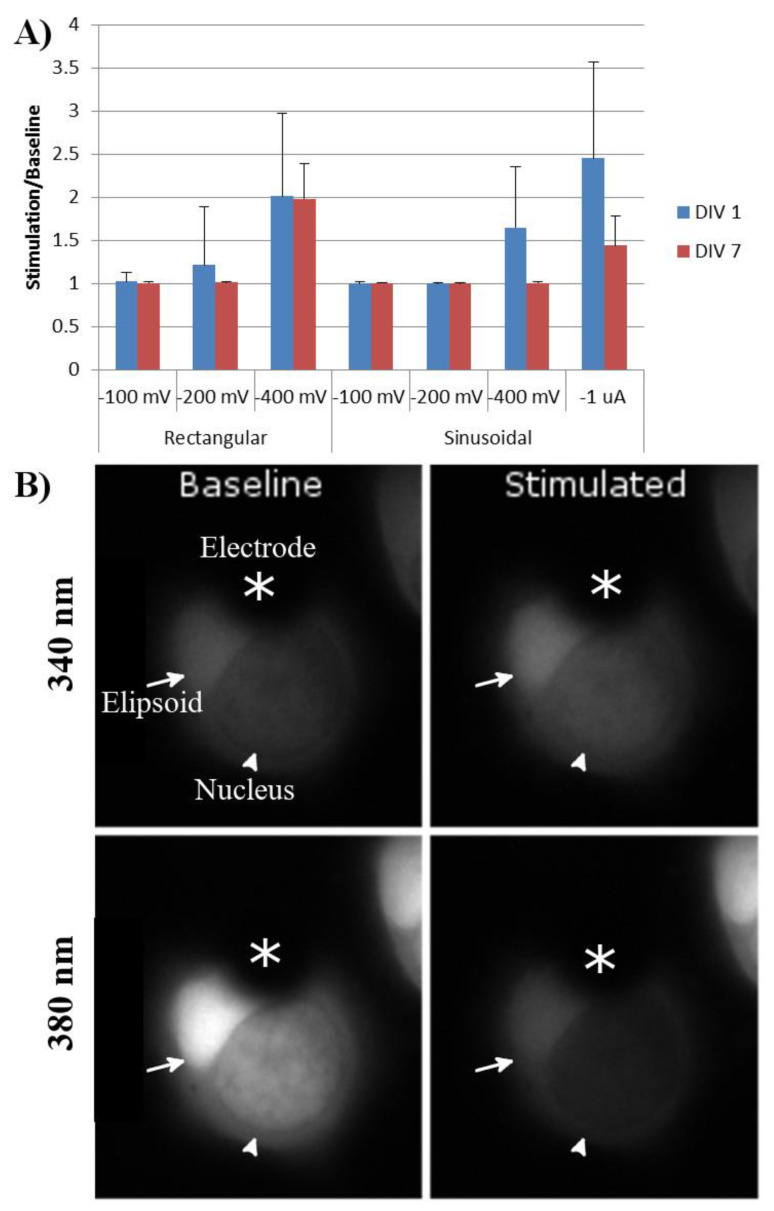
(**A**) Calcium response of tweezed rod photoreceptors to stimulation parameters of varying amplitude and type. Baseline was defined as the average intensity of 340 nm/380 nm for 50 s prior to stimulation. Response was defined as the average intensity of 340 nm/380 nm during stimulation. Rectangular stimulation was run with negative stimulation at the indicated amplitude for 100 µs, followed by no stimulation for 60 µs, followed by positive stimulation at the indicated amplitude for 100 µs. Sinusoidal stimulations were run at 5 Hz at the indicated amplitudes. *n* = 20 cells, four animals. Error bars = standard deviation. (**B**) Example of a rod photoreceptor one DIV prior to and during stimulation by the MEA electrode. Dark area covering a portion of the cell was the MEA electrode (asterisks). Bright area was the rod ellipsoid (arrow), while the other darker region was the nucleus of the rod photoreceptor (arrowhead).

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
