# Peer review of "Creating Custom Neural Circuits on Multiple Electrode Arrays Utilizing Optical Tweezers for Precise Nerve Cell Placement"

_mps, 2020, doi:10.3390/mps3020044_

Round 1

Reviewer 1 Report

Kung et al. have presented an interesting integration of microscope-based optical tweezers to Multiple Electrode Arrays (MEA) for precise nerve cell placement. The application of optical gradient forces to trap and reposition trapped objects dates back to the invention of optical tweezers technique by Arthur Ashkin (Ref. 7 in the manuscript). Recent examples can also easily be found, such as Ref [1] in this comment, where trapped PS beads are moved between two locations in a flow cell for different operations. That being said, the integration of optical tweezers to a MEA system is not seen before. The current work is novel in this sense. The current work gives detailed protocols of the integration of free-space optical tweezers to a MEA system. This would be a useful reference for researchers interested in related studies. In recent years, fast development of nanophotonics has enabled on-chip trapping and transport of objects via nanophotonic tweezers (Ref [2]). Looking ahead, it would be exciting if we can integrate the nanophotonic tweezers onto the MEA chip, so that neuron trapping and relocation can be done entirely on the MEA chip, without the need of the bulky optical microscope. Overall, the work is well written, with conclusions well supported. I would thus recommend acceptance of the manuscript upon addressing my following concern.

In the “2.4 Optical tweezing” section, the authors should provide more information on the focal depth of the 40x oil immersion objective used, as this would place some dimension limitation on the flow cell depth of MEA chamber.

[1] Killian, J.L.; Inman, J.T.; and Wang, M.D. High-Performance Image-Based Measurements of Biological Forces and Interactions in a Dual Optical Trap. ACS Nano 2018, 12, 11963-11974.

[2] Ye, F.; Badman, R.P.; Inman, J.T.; Soltani, M.; Killian, J.L.; and Wang, M.D. Biocompatible and High Stiffness Nanophotonic Trap Array for Precise and Versatile Manipulation. Nano Letters 2016, 16, 6661-6667.

Author Response

Thank you for your review of our manuscript.  We have addressed your comments in blue text below.

Kung et al. have presented an interesting integration of microscope-based optical tweezers to Multiple Electrode Arrays (MEA) for precise nerve cell placement. The application of optical gradient forces to trap and reposition trapped objects dates back to the invention of optical tweezers technique by Arthur Ashkin (Ref. 7 in the manuscript). Recent examples can also easily be found, such as Ref [1] in this comment, where trapped PS beads are moved between two locations in a flow cell for different operations. That being said, the integration of optical tweezers to a MEA system is not seen before. The current work is novel in this sense. The current work gives detailed protocols of the integration of free-space optical tweezers to a MEA system. This would be a useful reference for researchers interested in related studies. In recent years, fast development of nanophotonics has enabled on-chip trapping and transport of objects via nanophotonic tweezers (Ref [2]). Looking ahead, it would be exciting if we can integrate the nanophotonic tweezers onto the MEA chip, so that neuron trapping and relocation can be done entirely on the MEA chip, without the need of the bulky optical microscope. Overall, the work is well written, with conclusions well supported. I would thus recommend acceptance of the manuscript upon addressing my following concern.

In the “2.4 Optical tweezing” section, the authors should provide more information on the focal depth of the 40x oil immersion objective used, as this would place some dimension limitation on the flow cell depth of MEA chamber.

Thank you for the suggested additional detail that we should provide.  The focal depth or free working distance of the objective is 210 µm. We have added this information to the manuscript in subsection 2.3 line 113.

In addition, thank you for highlighting the development of nanophotonics for optical trapping.  We agree that it would be exciting to integrate nanophotonic tweezers onto the MEA chip so that trapping and relocation could be done entirely on the MEA without the need for a bulky optical tweezers assembly.  We have added the possibility of the use of nanophotonic tweezer arrays to the Discussion section line 372.

[1] Killian, J.L.; Inman, J.T.; and Wang, M.D. High-Performance Image-Based Measurements of Biological Forces and Interactions in a Dual Optical Trap. ACS Nano 2018, 12, 11963-11974.

[2] Ye, F.; Badman, R.P.; Inman, J.T.; Soltani, M.; Killian, J.L.; and Wang, M.D. Biocompatible and High Stiffness Nanophotonic Trap Array for Precise and Versatile Manipulation. Nano Letters2016, 16, 6661-6667.

Reviewer 2 Report

The manuscript: “Creating Custom Neural Circuits on Multiple Electrode Arrays Utilizing Optical Tweezers for Precise Nerve Cell Placement” by Frank H Kung  and Ellen Townes-Anderson proposes a method to precisely control placement of different types of neuronal retinal cells onto a multiple electrode array (MEA) using optical tweezers. This study expands a previously demonstrated optical tweezers protocol to physically manipulate individual cells and allow electrophysiological approaches to the study of neural networks. Optical manipulation of the cells and electrophysiological activity are confirmed by stimulating the photoreceptors with the MEA and measuring their response with calcium imaging.

Although optical trapping and manipulation of single cells is not new, its application to neuronal retinal cells in conjunction with a MEA device and calcium imaging can be considered as a (relatively) novel method. This work might be interesting for neurobiologists and help to implement optical manipulation in electrophysiology. The method is clearly presented and sustained by results. Therefore, I recommend this manuscript for publication pending to major revision, as implied by the comments below.

  1. In Introduction, line 55, it is claimed: “Using optical tweezers protocols developed by us, one group was able to place individual embryonic hippocampal granule cells onto a custom MEA with a “neurocage” design.[11] “. To my knowledge, there is no use of OT in ref [11]. The abstract says: “This micro-machined device consists of an array of neurocages which mechanically trap a neuron near an extracellular electrode.” Then, in section 1.2. The Neurochip of ref [11], the authors just mention that optical tweezers can be used to load cells in neurocages, citing another work (Chow G. Ph.D. thesis, Caltech. 2007. Laser Tweezers for Moving Live Dissociated Neurons. ): “the new neurochip is compatible with fabrication on a glass substrate (instead of silicon) which has significant implications for using optical tweezers to load the neurocages of devices with a large number of electrodes (Chow, 2007).” So, the authors of the present manuscript refer maybe to the work by Chow? If so, it should be cited. Please clarify. Then, please also discuss the novelty introduced by the present study with respect to the work above (beside the thesis there is also at least one paper by Chow).
  2. In subsection 2.3, line 94, it is mentioned that “thin MEAs are used due to optics of the objective…” Considering that the working distance WD of a microscope objective used in OT is reduced, usually about 300 um, this is an important piece of information. So please specify how thin are the MEAs you used? Are they commercially available or you thin them? Which is the WD and NA of the objective? What type of immersion? Although some information (NA, immersion) is mentioned later in the text, it should be introduced already here, in context of the MEA thickness. From the method point of view, all this info is important and should be provided to make it useful and reproducible for the reader.
  3. Cells identification (in 2.4. line 164) is just sent to ref [19] and then discussed to some etxtent at the beginning of results (3.1). I have two objections here. Cells identification is an important issue and it should be part of methods section. Identification, as far as I understand, is based by authors exclusively on cell morphology. A more careful and rigorous (quantitative) characterization is requested, using numbers for size and shape, plus definition of clear discrimination criteria.
  4. What happens with the cells which are not successfully placed on electrodes? There are still 25% cells not well placed.
  5. How do you define if a cell is successfully placed? It should touch the electrode with x % of its base surface? where the value of x should be defined according to a criterium. Please clarify.
  6. Does the electrode influence the manipulation? Diffraction, absorption or scattering of the trapping laser by the electrode do not influence the trapping and manipulation? Please comment. This issue is related to commnet 5.
  7. The displacement length and velocity are quite impressive and reduce the placement time. However, one has to consider that some time is lost to find and trap the cell, lift it up, before displacement and then lower it down to the electrode. Are these manipulation steps time consuming with respect to the displacement ? Please comment.
  8. Also speaking about velocity and throughput, could the authors comment on the use of multiple OT? Does it help ? Are there such studies already published ? Advantages, drawbacks ?
  9. Scale bars are missing in Figures 4 and 5.
  10. The fact that cells without a spherical shape can be manipulated by OT is not new (as it might appear from Discussion line 332). It was first demonstrated by Ashkin in his pioneering works between 1986 to 1989, and then shown also by many others. I recommend eliminating the ambiguity and adding at least one more citation by Ashkin (beside ref [7]).

Author Response

Thank you for your review of our manuscript.  We have provided our responses in blue text below.

  1. In Introduction, line 55, it is claimed: “Using optical tweezers protocols developed by us, one group was able to place individual embryonic hippocampal granule cells onto a custom MEA with a “neurocage” design.[11] “. To my knowledge, there is no use of OT in ref [11]. The abstract says: “This micro-machined device consists of an array of neurocages which mechanically trap a neuron near an extracellular electrode.” Then, in section 1.2. The Neurochip of ref [11], the authors just mention that optical tweezers can be used to load cells in neurocages, citing another work (Chow G. Ph.D. thesis, Caltech. 2007. Laser Tweezers for Moving Live Dissociated Neurons. ): “the new neurochip is compatible with fabrication on a glass substrate (instead of silicon) which has significant implications for using optical tweezers to load the neurocages of devices with a large number of electrodes (Chow, 2007).” So, the authors of the present manuscript refer maybe to the work by Chow? If so, it should be cited. Please clarify. Then, please also discuss the novelty introduced by the present study with respect to the work above (beside the thesis there is also at least one paper by Chow).

Thank you for addressing this issue.  We realize a more appropriate reference would be:
Pine, J.; Chow, G. Moving live dissociated neurons with an optical tweezer. IEEE Transactions on Biomedical Engineering 2009, 56(4): 1184-1188.

We have therefore replaced reference [11] with the reference noted above.

This work demonstrated optical tweezing onto a custom microelectrode array using the technique we developed for non-adherent substrates (Townes-Anderson et al 1998) which they cite.  Dr. Pine visited our lab to discuss our methods. Our work further expands upon Dr. Pine’s and Dr. Chow’s work by demonstrating the utility of optical tweezers in conjunction with a commercially available MEA and thus allowing biomedical researchers to more readily adapt this technique to their own laboratory.  

2. In subsection 2.3, line 94, it is mentioned that “thin MEAs are used due to optics of the objective…” Considering that the working distance WD of a microscope objective used in OT is reduced, usually about 300 um, this is an important piece of information. So please specify how thin are the MEAs you used? Are they commercially available or you thin them? Which is the WD and NA of the objective? What type of immersion? Although some information (NA, immersion) is mentioned later in the text, it should be introduced already here, in context of the MEA thickness. From the method point of view, all this info is important and should be provided to make it useful and reproducible for the reader.

The information on the MEAs in included in subsection 2.3.  However, we agree that additional detail regarding the MEA thickness should be included.  These were commercially available MEAs manufactured on #1 cover glass, thus about 0.17 mm thick. We have clarified that no physical alterations to these MEAs were required, subsection 2.3 line 114.

We have also provided additional information on the objectives used in subsection 2.3 on line 113.

3. Cells identification (in 2.4. line 164) is just sent to ref [19] and then discussed to some etxtent at the beginning of results (3.1). I have two objections here. Cells identification is an important issue and it should be part of methods section. Identification, as far as I understand, is based by authors exclusively on cell morphology. A more careful and rigorous (quantitative) characterization is requested, using numbers for size and shape, plus definition of clear discrimination criteria.

We agree that cell identification should be moved to the methods section and more detail should be provided. The new section starts on line 86. We have also added some additional information to Figure 4 to illustrate distinct cell features.

4. What happens with the cells which are not successfully placed on electrodes? There are still 25% cells not well placed.

Thank you for identifying a missing piece of information.  Cells that were not successfully placed were “lost” due to the movement of the cell through the cell medium.  It was unclear what eventually happened to these cells and whether they remained viable or not.  We have added this relevant information to subsection 3.1 line 243.

5. How do you define if a cell is successfully placed? It should touch the electrode with x % of its base surface? where the value of x should be defined according to a criterium. Please clarify.

Success was defined in subsection 2.4 line 194.  We have also added this to subsection 3.1 line 238 for additional clarity and for the reader’s convenience. In addition, we have defined success as coverage of a third or more of the electrode surface.

6. Does the electrode influence the manipulation? Diffraction, absorption or scattering of the trapping laser by the electrode do not influence the trapping and manipulation? Please comment. This issue is related to commnet 5.

The laser beam was likely absorbed by the electrodes and the optically clear tracks.  At high power and when directly focused on either the electrodes or the tracks, the laser could visibly damage both.  This detail has been added to subsection 2.4 line 192. 

The video of cell movement in the supplemental materials shows that cells were moved in the space between the electrode tracks. 

7. The displacement length and velocity are quite impressive and reduce the placement time. However, one has to consider that some time is lost to find and trap the cell, lift it up, before displacement and then lower it down to the electrode. Are these manipulation steps time consuming with respect to the displacement ? Please comment.

Identification of the correct cell type needed could be time-consuming for less experienced operators but improved with practice.

The lifting up of cells prior to movement and placing of cells onto the electrode after optical tweezing was a relatively quick process and took only ~30 seconds.

These details have been added to subsection 3.1 line 247.

8. Also speaking about velocity and throughput, could the authors comment on the use of multiple OT? Does it help ? Are there such studies already published ? Advantages, drawbacks ?

Multiple optical tweezer arrays have been utilized to manipulate other cell types and may be able to further increase the speed of the placement of cells onto the electrodes.  However, it may be difficult to accurately identify and trap cells based on morphology using a multiple optical tweezer array.  We have added this to the discussion on line 372 and included with a reference citing a multiple optical tweezer array and nanophotonic optical tweezer array for the trapping and movement of multiple cells.

9. Scale bars are missing in Figures 4 and 5.

We have added scale bars to figures 4 and 5.  Figure legends have also been updated in Figures 4 and 5.

10. The fact that cells without a spherical shape can be manipulated by OT is not new (as it might appear from Discussion line 332). It was first demonstrated by Ashkin in his pioneering works between 1986 to 1989, and then shown also by many others. I recommend eliminating the ambiguity and adding at least one more citation by Ashkin (beside ref [7]).

We agree and have added an additional reference to the work by Ashkin to demonstrate that this is not a novel finding. This addition can be found in the discussion to line 349.